# Connecting Obstetric, Maternity, Pediatric and Preventive Child Health Care: A Comparative Prospective Study Protocol

**DOI:** 10.3390/ijerph19116774

**Published:** 2022-06-01

**Authors:** Silke Boertien, Arie Franx, Danielle E. M. C. Jansen, Henk Akkermans, Marlou L. A. de Kroon

**Affiliations:** 1Department of Obstetrics and Gynecology, Erasmus MC—Sophia Children’s Hospital, 3015 CN Rotterdam, The Netherlands; a.franx@erasmusmc.nl; 2Department of General Practice and Elderly Care Medicine, University Medical Center Groningen, University of Groningen, 9712 CP Groningen, The Netherlands; d.e.m.c.jansen@umcg.nl; 3Department of Management, Tilburg University, 5037 AB Tilburg, The Netherlands; ha@tilburguniversity.edu; 4Department of Health Sciences, University Medical Center Groningen, University of Groningen, 9712 CP Groningen, The Netherlands; 5Department of Public Health and Primary Care, Centre for Environment and Health, Catholic University Leuven, 3000 Leuven, Belgium

**Keywords:** integrated care, preventive child health care, obstetric care, first 1000 days, collaboration in care, interprofessional relations, interorganizational relations

## Abstract

Collaboration between birth care and Preventive Child Health Care (PCHC) in the Netherlands is so far insufficient. The aim of the Connecting Obstetric; Maternity; Pediatric and PCHC (COMPLETE) study is to: (1) better understand the collaboration between birth care and PCHC and its underlying mechanisms (including barriers and facilitators); (2) investigate whether a new multidisciplinary strategy that is developed as part of the project will result in improved collaboration. To realize the first aim, a mixed-method study composed of a (focus group) interview study, a multiple case study and a survey study will be conducted. To realize the second aim, the new strategy will be piloted in two regions in an iterative process to evaluate and refine it, following the Participatory Action Research (PAR) approach. A prospective study will be conducted to compare outcomes related to child health, patient reported outcomes and experiences and quality of care between three different cohorts (i.e., those that were recruited before, during and after the implementation of the strategy). With our study we wish to contribute to a better understanding of collaboration in care and develop knowledge on how the integration of birth care and PCHC is envisioned by stakeholders, as well as how it can be translated into practice.

## 1. Introduction

A person’s first 1000 days affect their health and opportunities throughout life [1]. During this uniquely relevant period from conception to a child’s second birthday, the foundations of a child’s health, growth and neurodevelopment are established [2]. Therefore, the care that infants receive within the first 1000 days of their lives is of vital importance [3]. In the Netherlands, several domains are involved in the delivery of care during this crucial period. Among these are midwifery care, obstetric care, maternity care, neonatal/pediatric care, family medicine and Preventive Child Health Care (PCHC), including both medical specialist and allied healthcare professionals. It is hypothesized that integrating the care these parties provide is one of the ways in which perinatal health outcomes, and subsequently the lifelong health and opportunities of the child, can be improved [4]. The Dutch government and the main parties within the sector embrace this hypothesis [5,6]. The College of Perinatal Healthcare, the national overarching birth care network-organization states that it is their aim to provide ‘state of the art integrated care’ by 2022 [5]. One of the ways to realize this aim has been the introduction of the ‘Care Standard Integrated Birth Care’ (CSIBC) in 2016. The CSIBC has obligated birth care organizations to commit to the integration of birth care, including a close collaboration with PCHC.

This collaboration in particular has proven to be challenging. Before the introduction of the CSIBC, the Inspectorate of Health Care and Youth (IHCY) presented two reports in 2014 and 2016 in which they assessed that collaboration was insufficiently realized and needed to be improved to promote both the continuity and quality of care in the first 1000 days [7,8]. Since the publication of these reports and the introduction of the CSIBC collaboration has been encouraged by the introduction of several local interventions requiring effective collaboration for their implementation, and by the national program ‘Solid Start’ (Kansrijke Start), launched by the Ministry of Health, Welfare and Sport in 2018. Despite these efforts, the National Institute of Public Health and the Environment (NIPHE) still identified collaboration between birth care and PCHC as ‘absent, or hesitantly realized’ [9] (p. 40) in their report ‘To know better: a better start’, which was published in December 2020. This conclusion is further substantiated by a recent qualitative study investigating the handover of care to PCHC [10].

Although the above-mentioned studies show a need for improvement of the collaborative relationship between birth care and PCHC in the Netherlands, it remains unclear how this can be achieved. Studies from other countries that are currently in the transition towards integrated birth care, such as studies from Sweden [11,12,13]; Australia [14,15,16,17,18,19]; Canada [20,21]; the UK [22,23]; and Norway [24] have already offered valuable insights into the collaborative process. However, as the organization of healthcare differs between countries, research within the Dutch context is needed to develop successful strategies for the Netherlands as well. The Connecting Obstetric, Maternity, Pediatric and Preventive Child Health Care (COMPLETE) study was designed to address this knowledge gap.

The aim of the COMPLETE study is twofold. The first aim is to gain more insight into the current collaborative relationship between birth care and PCHC in the Netherlands and its underlying mechanisms (including barriers and facilitators); the second aim is to develop and test a new, evidence-based multidisciplinary strategy to improve collaboration.

## 2. Materials and Methods

### 2.1. General Approach

The COMPLETE study is comprised of two subsequent steps. To realize our first research aim, namely, gaining more insight into the current collaborative relationship between birth care and PCHC, we will use a mixed methods approach to better understand current collaborative practices. This step will be described in further detail in Section 2.2. To realize our second aim, developing and testing a new strategy to improve collaboration, we will subsequently develop a strategy to improve collaboration based on the results from the first step. This strategy will be improved and refined during a pilot. This step will be described in further detail in Section 2.3. If the final strategy successfully improves collaboration between birth care and PCHC professionals, a plan for national implementation will be developed. Each sub-study within the COMPLETE project will be submitted for review by the MERC Erasmus MC, separately, and will only be conducted after their approval. All data that are collected within our study will be either pseudomized or anonymized and will be stored on a secure, password-protected location on an Erasmus MC server. The data will be destroyed after 15 years.

### 2.2. Mixed Method Study

The mixed method study will have a partially mixed concurrent equal status design and consists of: (1) a (focus group) interview study; (2) a multiple case study; (3) a survey study. The data from both qualitative studies and the quantitative study will be integrated during the interpretation stage. We have chosen an equal status design because we consider the three studies we will conduct to be complementary to each other: The (focus group) interview study allows us to explore experiences and views of stakeholders [25], the multiple case study helps us to better understand the process of collaboration [26], and with a quantitative descriptive design we can assess the prevalence level of collaboration throughout the Netherlands. The development of the new strategy will be based on the results from all three components of the mixed method study.

#### 2.2.1. The (Focus Group) Interview Study

We will use a qualitative (focus group) interview study design to explore how the collaboration between birth care and PCHC is experienced and viewed by the different stakeholders and identify possible facilitators and barriers. We will organize four focus groups with professionals, managers, policy makers and experts and 8–12 in-depth-interviews with parents. This sub-study is a nWMO study and has been separately approved by the Medical Ethics Review Committee (MERC) Erasmus MC.

##### Setting and Participants

The focus group discussions and interviews will be conducted online and recorded using Zoom due to the COVID-19 restrictions that are in place in the Netherlands. Participants are informed that they have a 5-day reflection period after the interview to decide whether they wish to withdraw or not. If they wish to withdraw from the study, their data will be destroyed. Withdrawing from the study has no further consequences. In each focus group discussion 6–8 participants will participate. The first two focus groups will consist of health care professionals. We will employ a purposive sampling method to ensure maximum variation with regard to professional background. In both focus groups with professionals, two PCHC professionals, one maternity care professional, one midwife specializing in low-risk pregnancies, one midwife specializing in high-risk pregnancies, one obstetrician, one neonatologist and one general practitioner will be invited to represent the different disciplines. We will invite professionals from different regions in the Netherlands, as the local level of the integration of birth care and PCHC varies greatly [27]. Professional background and geographical location will also inform our sampling strategy for the third and fourth focus group discussions, to which managers, policy makers and experts will be invited.

Potential participants for the interviews will be recruited through parent advisory boards from the Obstetric Care Networks (OCN’s), which are regionally organized multidisciplinary networks responsible for the local organization of obstetric care. We will also use our own network to find clients with a socially less advantaged background to ensure diversity in our data, as parents who are active in advisory boards more often have a relatively high socioeconomic status. Parents are eligible for the study if they: (1) have a child <3 years old; (2) are over 18 years old; (3) are willing to give written permission for videotaping the interview. Parents who insufficiently understand the Dutch language will not be able to participate.

##### Data Collection

The focus group interviews, to which the professionals will be invited, will be conducted by two researchers and will take 2 h each. A topic list based on the framework of D’Amour et al. [28] will be used to guide the discussion. This framework offers a way to analyze complex, heterogeneous multi-level systems which involve both interprofessional and interorganizational collaboration [29]. Within this framework the following themes regarding collaboration were identified: shared goals, mutual acquaintance, trust, roles and responsibilities, connectivity and accessibility, patient centeredness, information exchange, formalization tools and innovation. During the discussion the participants will be invited to discuss their experiences with and opinions about the following processes: coordination of care (including referrals), handover of care, sharing feedback, offering care pathways and the development of long-term projects.

The one-on-one interviews with the parents will be conducted by one researcher and will last a maximum of 1 h each. An interview guide will be used and include questions about: (1) personal characteristics, such as age, parity and education; (2) overall experience of pregnancy, birth and the postnatal period; (3) overall experience of prenatal care and postnatal care; (4) experience of the introduction to PCHC; (5) experience of continuity of care; (6) suggestions to improve care. To safeguard data security, we will use a paid Zoom-account, enable password protection during the video-calls and save the recordings locally to a secure, password-protected location [30,31,32]. Participants will be blocked from the opportunity to record during the Zoom-call.

##### Data Processing and Analysis

The (focus group) interviews will be transcribed verbatim, pseudonymized and returned to the participants for a member check. After the transcriptions are finalized, the recordings will immediately be destroyed. The transcripts will be coded in ATLAS.ti using the thematic analysis method as explicated by Braun and Clarke [33]. The coding process will be aimed at providing a detailed account of how the collaboration between birth care and PCHC is experienced and viewed. Other topics which might be discussed by the participants will not be considered in the analysis.

#### 2.2.2. Multiple Case Study

We will use a case study design to: (1) explore how the collaboration between different healthcare providers is experienced by the parents as well as the healthcare providers themselves; (2) map the collaborative process between these parties; (3) gain a better understanding of the difficulties and challenges of collaboration and service integration. The multiple case study design will allow us to compare outcomes across different settings and thus to identify the possible effects of different environments and specific conditions on the outcomes in individual cases [34]. This can help to identify factors that promote or inhibit effective collaboration. Our goal is to build a theoretical framework which explicates the mechanisms behind and challenges to the collaborative processes that take place between the healthcare providers that support the new or expanding family before and after birth.

We will use the World Health Organization health system building blocks to guide our study [35]. This conceptual framework helps to identify and understand the challenges to health systems and relate service delivery and health outcomes to the organization of care. The framework focuses explicitly on strengthening health systems. This sub-study is a nWMO study and was separately approved by the METC Erasmus MC.

##### Defining and Selecting Cases

In our multiple case-study, the pre- to postnatal care trajectory of the mother and child will be our main unit of analysis. The different interactions that occur throughout the care trajectory function as ‘sub-events’ within the cases.

We will solicit professionals from different disciplines for clear cases of poor or ineffective collaboration. Exploring these cases of process failure will help us to identify different types of problems in the collaborative process. Each case will need to meet the following criteria: (1) the case is recent (meaning the last contact between the care provider who introduces the case and the client took place <3 months ago); (2) the parents have given consent to their care provider to suggest their case for our research purposes; (3) the parents have indicated that at least one of them is willing to participate in an interview themselves.

Because we aim to compare cases with each other, we will investigate theoretically different case settings in which collaboration failed for different reasons. For instance, we will search for cases in which the parents and children we include followed different care pathways. Different care pathways require different healthcare providers to be involved. Thus, what hinders collaboration between them may also be different. Ex ante, we can already preselect several case settings that appear to have a higher-than-average likelihood of leading to problematic process outcomes, such as prematurity, mental health issues and financial issues. We will also select cases from a variety of locations, as this will allow us to better identify the effects of different socio-geographical environments. Our case selection process will be iterative; the results from the first cases will inform who we will ask for new cases and what kind of cases we will look for.

##### Recruitment and Data Collection

To explore the different perspectives on the care trajectories that comprise our cases, the stakeholders (such as the mother, midwife or PCHC nurse) involved with the selected cases will be invited for an interview separately. When a case is selected, we will first interview (one of) the parents and the care provider who initially presented the case to us. We will then approach the other healthcare providers.

The interviews will take place online using Zoom, or face-to-face if this is possible, despite the COVID-19 pandemic. The same security measures with regard to the use of Zoom as described above will be taken. The interviews will be guided by an interview guide and will take approximately 60 min. During the interviews, the participants will be asked to elaborate on the different interactions that took place during the care trajectory. For this study we are interested in how our participants describe these interactions, how they have experienced them and how our participants were affected by them. The interview data will be further triangulated and substantiated by an analysis of the medical records of the mother and child. Before the interviews take place, professionals will be informed that they have a 5-day reflection period after the interview to decide whether their wish to withdraw from the interview. If they wish to withdraw from the study, only the data that were collected prior to participant withdrawal will be used in our study. Withdrawing from the study has no further consequences. The parents can also withdraw their participation after the 5-day reflection period; if they wish to do so, all data related to their case will be destroyed and the case will not be investigated further.

To conduct this study, we need cooperation and consent from both the parents and different professionals. We anticipate that it may be difficult to include the care trajectories of mothers in a socially disadvantaged situation. These mothers are less likely to participate as they are often harder to reach [36]. To overcome this issue, we will specifically target networks of professionals specifically working with mothers from a low SES.

##### Data Processing and Analysis

The interviews will be transcribed verbatim, anonymized and returned to the participants for a member check. After the transcriptions are finalized, the recordings will be destroyed. The data will be coded using the Constant Comparison Analysis method, in ATLAS.ti. The goal will be to build a theoretical framework which explicates the mechanisms behind and challenges to the collaborative processes that take place between the healthcare providers that support the new or expanding family before and after birth.

#### 2.2.3. Survey Study

We will use a cross-sectional descriptive study design to help assess: (1) the prevalence of integrative practices; (2) the level of current integration of birth care and PCHC in the Netherlands. The survey study will be informed by the Development Model of Integrated Care (DMIC) which was developed by Minkman et al. [37] and will focus on two main topics. The first topic regards the birth care and PCHC professionals’ practices, identified as integrative activities in the DMIC. The second regards the level of integration, determined in accordance with the four phases of the integration process in the DMIC. The study will only be conducted after approval is given by the METC Erasmus MC.

##### Setting and Participants

Data will be collected from all 71 OCNs, all 9 integrated birth care organizations and all regional PCHC organizations (>90 in 2014 according to the Netherlands Centre Youth Health website). All professionals in these organizations will be asked to participate in the survey. However, as the composition of the OCNs varies, we will take care that—if needed—maternity care professionals and pediatricians will also be selected from organizations not belonging to the OCNs. Participants will be informed that submitting the questionnaire is taken as providing informed consent.

##### Sample Size Considerations

To assess the correlations between an outcome and several independent variables we need at least 10 cases for each independent variable, enabling multiple regression analyses. This implies that we will need at least 100 respondents that are representative of several disciplines to assess the relevant significant relationships between the outcomes and the 10 independent variables.

##### Data Collection

Data for this study will be collected using a survey that consists of three distinct components. The first component will be an instrument to measure the practices of birth care and PCHC professionals, based on the DMIC [37], consisting of 89 items (subdivided into nine dimensions) that correspond to activities which are considered integrative. For each item, the birth care and PCHC professionals will be asked to answer yes or no to questions that relate to: (1) the relevance of the activity to their practice; (2) its presence. The second component will be an instrument that can be used to determine the level of advancement of integration; it is a validated grid, developed by Minkman et al. [37]. This grid includes 40 activities (out of the 89 integrative activities) that are considered to be the most significantly representative of the four phases of the integration process (10 activities per phase). Respondents will be asked to mark which representative items associated to the different phases are prevalent to determine which level of integration their organization has reached. The third component will consist of a questionnaire to capture information on sociodemographic variables and construct a general profile of the respondents. Both DMIC instruments are available in Dutch and have been validated in various care pathways [37].

The online survey applications Limesurvey and Gemstracker will be used to collect the survey data. The survey will be online for a month.

##### Processing and Analysis

To address our first objective, we will use descriptive statistics to examine both the relevance and presence of the 89 activities, expressed in percentages and averages. In accordance with the DMIC, we will consider an activity to be relevant and/or present when at least 60% (≥60%) of respondents have answered ‘yes’ to the corresponding item [38]. Only if an activity is deemed relevant will its presence be considered. When 60% or more of the items associated with one of the nine dimensions are present, this dimension will be considered prevalent. To address our second objective, descriptive statistics will be used to determine the level of integration, expressed in percentages and averages. In accordance with the DMIC, a given phase will be reached when the score of activities which are considered significant for that specific phase is equal to or greater than 7 out of 10 activities (score ≥ 70%) [39]. To produce respondent profiles and to examine the associations between perceptions of activity presence and different variables, including the respondent’s discipline and practice location, both descriptive and variance analyses will be conducted.

### 2.3. The Complete Pilot

Based on the results of the mixed method study we will develop a new strategy to improve collaboration. The strategy will be tested for one year and five months to evaluate and refine the strategy through an iterative process. To determine whether the new strategy successfully improves collaboration between birth care and PCHC in the pilot regions, a prospective study will be conducted that compares outcomes in three cohorts, i.e., those recruited before, during and after the implementation of the strategy. To determine the effectiveness of the strategy we will measure: (1) child health outcomes (captured by PCHC professionals during the first 6 months of life); (2) value-based healthcare outcomes representing experiences of clients as developed by the International Consortium for Health Outcomes Measurement (ICHOM); (3) the experiences of professionals. The pilot will only start after approval of the MERC Erasmus MC.

#### 2.3.1. Setting and Participants

The new strategy will be piloted in two OCNs and the PCHC organizations in the corresponding regions. We will take the following into account: first, we will include regions which are theoretically different; second, we will choose pilot regions in which both OCN and PCHC organizations participate in the VIPP program, Babyconnect. This program aims at the realization of digital data transfer in the Netherlands, a significant precondition for effective collaboration. Out of the 71 OCNs and 9 integrated birth care organizations 45 already participated in the program and the other 26 OCNs plan to do so in the future [40].

#### 2.3.2. Implementation of the Strategy

We aim to refine, evaluate and re-evaluate our strategy during the pilot through an iterative process. During this process we will actively involve the organizations, professionals and client-representation in the region and conduct our research following the principles of Participatory Action Research (PAR). PAR encompasses a “process of fact finding, action, reflection, leading to further inquiry and action for change” [41] (p. 37), in which the involvement and empowerment of stakeholders and an equal relationship between researchers and respondents is fostered [41].

#### 2.3.3. Data Collection Process

During the pilot we will collect data to evaluate and refine our strategy using surveys, focus group interviews and data from client files. We use the ACTion procedure to guide our data collection [42]. This procedure consists of three steps, namely, problem analysis (or baseline assessment), improvement and evaluation. Within the context of the COMPLETE study this means we will: (1) perform a baseline assessment of the outcome measures and risk factors (see below); (2) conduct follow-up assessments during the pilot, using the same outcomes, with the aim of improving the strategy; (3) organize focus groups with all stakeholders to agree on adaptations of the strategy; (4) conduct a final assessment of the outcomes in relation to risk factors and finalize the strategy. Parents who wish to participate in the pilot will be asked for their informed consent in writing and will be informed that they can withdraw from the study at any time, without any consequences. Their data will be removed from the study database.

#### 2.3.4. Outcome Measures

To assess value-based health care outcomes we will use several instruments from the ICHOM Pregnancy and Birth standard outcome set [43], namely, the Patient-Reported Outcomes Measurement Information System Global 10 [44] to assess health-related quality of life; the Breastfeeding Self-Efficacy Scale-Short Form [45] to assess confidence with breastfeeding; the Mother-Infant Bonding Scale [46] to assess mother-infant attachment; the Patient Health Questionnaire 2 [47] and Edinburgh Postnatal Depression Scale [48] to assess post-partum depression; and the Birth Satisfaction Scale-Revised [49] to assess birth experience.

The ICHOM Pregnancy and Birth standard outcome set was developed with the aim of measuring client outcomes which matter the most to pregnant and laboring women, from early pregnancy to six months post-partum [43]. It has been translated and validated for use in the Netherlands [50] and includes a list of case-mix factors which allows for comparison across and between patient populations. The outcomes will be measured during five moments: the first prenatal consultation in the first trimester; the early third trimester (between 28–32 gestational age); the discharge from the birth unit or three days post-partum; six weeks post-partum; and six months post-partum. This last measurement gives an indication of the long-term effects of the strategy.

Child health outcomes will be assessed using the National Obstetric Registration and the digitized files of the PCHC and will include: (1) gestational age and being small-for-gestational age as predictors of growth and development [51,52]; (2) the APGAR score; (3) growth until the age of six months by assessment of length and weight [52]; (4) developmental milestones assessed with the Van Wiechen Instrument [53] until the age of 6 months.

Quality of care will be measured using surveys taken by both clients and professionals, including questions on: (1) collaborative behavior; (2) possible advantages and disadvantages of the strategy; (3) knowledge, skills and attitudes regarding collaboration; (4) points for improvement; (5) perceived quality of care by clients, also assessed with a selection of the ICHOM instruments.

#### 2.3.5. Independent Variable

The independent variable will be the introduction of the new strategy (categorized as ‘no’; ‘yes, the first version of the strategy’; ‘yes, the second version of the strategy’; and ‘yes, the final version of the strategy’).

#### 2.3.6. Statistical Analyses

The statistical analyses that will be performed during the baseline assessment, later assessments and final assessment will include descriptive analyses, t-tests, chi-square tests and non-parametric tests (where applicable) to describe the characteristics of the populations at baseline, during the assessments, during the pilot and at the end of the pilot, regarding age; parity; gender of the child; and socioeconomic status (SES), which is an important predictor of birth and pregnancy outcomes. Next, we will carry out linear univariable and multivariable regression analyses to assess the differences between the outcomes measured, before, after the initial introduction of the strategy, and after adaptations of the strategy in the organizations that participate in the pilot. Several variables will be included as covariates (including the child’s age, parity, ethnicity and SES).

#### 2.3.7. Sample Size Calculation

The power analysis for the number of mothers (and their infants) to be included is based on health outcomes before and after the introduction of the strategy. For the power analysis we consider QOL z-scores in this population as indicated by the mothers in the ICHOM questionnaire as the main outcome, because this outcome is strongly related to the other health outcomes (of mothers and children). To be able to detect an increase in z-score with 0.5 (and a SD of 1) after the introduction of the final strategy, with a power of 80% and an alpha of 0.05, we will have to include at least 63 women (and their infants) at baseline, the second assessment and the final assessment for each OCN-PCHC collaboration (a total of 189 women at the initial, the second and the final assessment).

The power analysis of the number of OCN-PCHC collaborations to be included is based on questions, categorized according to a 5-points Likert-scale. To be able to detect an increase in the Likert-scale with 1 (between the baseline assessment and the final assessment) and an SD of 1.2 (on a 5-points Likert-scale), with a power of 80% and an alpha of 0.05, we will have to include at least 23 professionals of both OCNs and PCHC organizations (a total of 46 professionals).

## 3. Conclusions

The care that infants receive during their first 1000 days after conception may strongly affect their health and opportunities throughout life [1,2]. The integration of healthcare during these pivotal years is hypothesized to positively impact child health [4]; thus, there has been an increased interest in efforts to realize this throughout the past decade [54]. Investing in excellent care in the first 1000 days can be considered as an ultimate expression of preventive health care; it is an investment in the health of future generations. Moreover, such an investment can help mitigate the growth of health care costs in the future [55]. Additionally, collaboration in care promotes the experience of continuity of care [13,56,57], which is highly valued by parents and can help them in their transition towards parenthood [58,59]. Professionals can benefit too, as effective collaboration can promote enjoyment of their work [29]. The COMPLETE study will advance our understanding of interprofessional and interorganizational collaboration and integration of care and contribute to the realization of expected benefits of collaboration for parents and their children and for involved professionals. We expect that the results of the first step of our study, the mixed method study, will contribute to a better understanding of collaboration between birth care and PCHC from the perspectives of various stakeholders, the underlying mechanisms of collaboration (including the barriers and facilitators) and to our knowledge of interprofessional and interorganizational collaborative processes in pre-, peri- and postnatal care, specifically within the Dutch context. We expect that the results of the second step of our study, the pilot of the new multidisciplinary strategy, will help to explore how a complex heterogeneous collaboration such as the collaboration between birth care and PCHC can be successfully optimized, and how a strategy targeting collaboration can be implemented. The pilot can also help us to better understand the unforeseen effects that might arise from implementing a new strategy targeting collaboration, and offer new insights relating to the field of implementation research.

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
