# Peer review of "Connecting Obstetric, Maternity, Pediatric and Preventive Child Health Care: A Comparative Prospective Study Protocol"

_ijerph, 2022, doi:10.3390/ijerph19116774_

Round 1
Reviewer 1 Report
Dear authors,
It is an interesting topic. However there are some suggestions:
Introduction: You need to identify the ethical approval.
Methods: clarify the topics for each aim.
Results: The results are a little confused, maybe try another way to divide them. You must always use different aims in some form.
Discussion: You need to have a topic with the discussion of the results with others references.
Best regards
Author Response
Rotterdam, May 16th 2022
Dear reviewer,
Thank you very much for reviewing our manuscript. Below you will find our responses to your comments and the changes we have made to our manuscript.
List of remarks of the reviewer and our responses (responses are in italics):
1) Introduction: You need to identify the ethical approval.
Thank you for this suggestion. In the original text the ethical approval of the different sub-studies were described in lines 113-114,192-193, 262 and lines 326-327. The ethical approval was also described, i.e. in the Institutional Review Board Statement (lines 457-458). To describe the ethical approval earlier in our manuscript, as suggested by the reviewer, we have now added the following sentence:
“Each sub-study within the COMPLETE project will be submitted for review by the MERC Erasmus MC separately and will only be conducted after their approval.” (lines 92-93)
2) Methods: clarify the topics for each aim.
We have edited the general approach section to clarify each aim. The section was edited to:
“The COMPLETE study is comprised of two subsequent steps. To realize our first research aim, namely, gaining more insight into the current collaborative relationship between birth care and PCHC, we will use a mixed methods approach to better understand current collaborative practices. This step will be described in further detail in section 2.2. To realize our second aim, developing and testing a new strategy to improve collaboration, we will subsequently develop a strategy to improve collaboration based on the results from the first step. This strategy will be improved and refined during a pilot. This step will be described in further detail in section 2.3. If the final strategy successfully improves collaboration between birth care and PCHC professionals a plan for national implementation will be developed.” (lines 82-91)
3) Results: The results are a little confused, maybe try another way to divide them. You must always use different aims in some form.
Thank you for your suggestion.. Since this is a paper on a study protocol, we cannot yet describe results. However, in line with your feedback we have rewritten the text on the expected results in the Conclusion section analogous to the two aims of the study, as follows:
“ The care infants receive during their first 1000 days after conception may strongly affect their health and opportunities throughout life [1,2]. The integration of healthcare during these pivotal years is hypothesized to positively impact child health [4]. The past decade there has therefore been an increased interest in efforts to realize this [55]. Investing in excellent care in the first 1000 days can be considered as an ultimate expression of preventive health care: it is an investment in the health of future generations. Moreover, such an investment can help mitigate the growth of health care costs in the future [56]. Additionally, collaboration in care promotes the experience of continuity of care [13, 57, 58], which is highly valued by parents and can help them in their transition towards parenthood [59, 60]. Professionals can benefit too, as effective collaboration can promote the enjoyment of their work [30].
The COMPLETE study will advance our understanding of interprofessional and interorganizational collaboration and integration of care and contribute to the realization of expected benefits of collaboration for parents and their children and for involved professionals. We expect that the results of the first step of our study, the mixed method study, will contribute to a better understanding of collaboration between birth care and PCHC from the perspectives of various stakeholders, the underlying mechanisms of collaboration (including the barriers and facilitators) and to our knowledge of interprofessional and interorganizational collaborative processes in pre-, peri- and postnatal care, specifically within the Dutch context.
We expect that the results of the second step of our study, the pilot of the new multidisciplinary strategy, will help explore how a complex, heterogeneous, collaboration such as the collaboration between birth care and PCHC, can be successfully optimized and how a strategy targeting collaboration can be implemented. The pilot can also help us better understand unforeseen effects that might arise from implementing a new strategy targeting collaboration and offer new insights relating to the field of implementation research.” (lines 426-451).
4) Discussion: You need to have a topic with the discussion of the results with others references.
Thank you for your suggestion. In line with the aim of a design article, in which we do not have any results yet, we are not able to discuss our results in relation to other academic work. However, also in line with your earlier remark we have edited our conclusion and now more explicitly discuss the expected results and implications of our research (see above).
Thank you for your attention, also on behalf of the co-authors,
Silke Boertien, MSc
Reviewer 2 Report
In this study Protocol, Silke Boertien et al. analyzed the relationship between birth care and Preventive Child Health Care (PCHC) emphasizing barriers or facilitators to improve the collaboration. The manuscript is generally well written but in my opinion it needs minor revision. My concerns are listed below:
- (Lines 81-96) The two subheadings can be unified to facilitate understanding of the study method.
- (Lines 130-141) Did the interview last an hour or two?
Author Response
Rotterdam, May 16th 2022
Dear reviewer,
Thank you very much for reviewing our manuscript. Below you will find our responses to your comments and the changes we have made to our manuscript.
List of remarks of the reviewer and our responses (responses are in italics):
1) (Lines 81-96) The two subheadings can be unified to facilitate understanding of the study method.
Thank you for your suggestion. From your remark it has become clear to us that we should have communicated that the mixed-method study and pilot study are separate, subsequent, steps within our general approach. We have therefore edited this section by adding the following information:
“The COMPLETE study is comprised of two subsequent steps. To realize our first research aim, namely, gaining more insight into the current collaborative relationship between birth care and PCHC, we will use a mixed methods approach to better understand current collaborative practices. This step will be described in further detail in section 2.2. To realize our second aim, developing and testing a new strategy to improve collaboration, we will subsequently develop a strategy to improve collaboration based on the results from the first step. This strategy will be improved and refined during a pilot. This step will be described in further detail in section 2.3. If the final strategy successfully improves collaboration between birth care and PCHC professionals a plan for national implementation will be developed.” (lines 82-91)
2) (Lines 130-141) Did the interview last an hour or two?
From you remark it has become clear to us that we did not add this information in our paper. We have therefore made the following edits:
“The focus group interviews to which the professionals will be invited, will be conducted by two researchers and will take 2 hours each.” (lines 144-145)
“The interviews with the parents will be conducted by one researcher and will last a maximum of 1 hour each.” (lines 155-156)
Thank you for your attention, also on behalf of the co-authors,
Silke Boertien, MSc
Reviewer 3 Report
This is an excellent proposal. I have a few minor comments only, and I wish you every success when you begin data collection.
In your Abstract, second line (line 19 on page), change "is" to "has been".
Introduction: Line 42/43 can you please specifiy if "pediatric care" means therapies / allied healthcare professionals? Please do, as this will increase readership and interest to a wider group.
Page 5: Re: Zoom sessions. Please specify if sessions will be recoded, and how confidentiality will be maintained?
Also, please be clear about data storage. Can you add what will happen to data where participants decide to withdraw from the study?
Author Response
Rotterdam, May 16th 2022
Dear reviewer,
Thank you very much for reviewing our manuscript and your kind words. Below you will find our responses to your comments and the changes we have made to our manuscript.
List of remarks of the reviewer and our responses (responses are in italics):
1) (line 19 on page), change "is" to "has been"
Thank you for your suggestion. However, we would like to clarify to the reviewer that we are still conducting our study and have therefore used the present tense.
2) Line 42/43 can you please specify if "pediatric care" means therapies / allied healthcare professionals? Please do, as this will increase readership and interest to a wider group.
Thank you for your suggestion. To better specify if pediatric care also means allied healthcare professionals we have edited lines 42/43 to:
“Among these are midwifery care, obstetric care, maternity care, neonatal/ pediatric care, family medicine and Preventive Child Health Care (PCHC), including both medical specialist and allied healthcare professionals.” (lines 42-44)
3) Zoom sessions. Please specify if sessions will be recoded, and how confidentiality will be maintained?
Thank you for your suggestion. To specify how confidentiality will be maintained we have added the following sentences:
“To safeguard data security we will use a payed Zoom-account, enable password protection during the video-calls and safe the recordings locally to a secure, password-protected location [31-33]. Other participants will be blocked from the opportunity to record during the Zoom-call.” (lines 162-165)
“The same security measures with regard to the use of Zoom as described above will be taken.” (lines 225-226)
4) Please be clear about data storage. Can you add what will happen to data where participants decide to withdraw from the study?
Thank you for your question. We have now expressed more elaborately how we will store our data and what will happen to the data when participants decide to withdraw by adding the following sentences:
“All data that is collected within our study will be either pseudomized or anonymized and will be stored on a secure, password-protected location on an Erasmus MC server. The will be destroyed after 15 years.” (lines 93-96)
“Participants are informed that they have a 5-day reflection period after the interview to decide whether they wish to withdraw or not. If they wish to withdraw from the study, their data will be destroyed. Withdrawing from the study has no further consequences.” (lines 118-121)
“After the transcriptions are finalized the recordings will immediately be destroyed.” (lines 169-170)
“Before the interviews take place professionals are informed that they have a 5-day reflection period after the interview to decide whether their wish to withdraw from the interview. If they wish to withdraw from the study, only the data collected that was collected prior to participant withdrawal will be used in our study. Withdrawing from the study has no further consequences. The parents can also withdraw their participation after the 5-day reflection period; if they wish to do so all data related to their case will be destroyed and the case will not be investigated further.” (lines 233-239)
“After the transcriptions are finalized the recordings will be destroyed.” (lines 248-249)
“Participants will be informed that submitting the questionnaire is taken as providing in-formed consent.” (lines 270-271)
“Parents who wish to participate in the pilot will be asked for their informed consent in writing and will be informed they can withdraw from the study at any time, without any consequences. Their data will be removed from the study database.” (lines 358-360)
Thank you for your attention, also on behalf of the co-authors,
Silke Boertien, MSc